# The Effect of Lithium Doping on the Dielectric Properties of Solid Solutions $Li_xCa_{(1-x)}Cu_3Ti_4O_{12}$ (x = 0.01–0.1)

Alexey Tsyganov *, Natalia Morozova, Maria Vikulova, Aleksandra Asmolova, Denis Artyukhov, Ilya Zotov, Alexander Gorokhovsky and Nikolay Gorshkov *

Department of Chemistry and Technology of Materials, Yuri Gagarin State Technical University of Saratov, 77 Polytecnicheskaya Street, 410054 Saratov, Russia

* Correspondence: tsyganov.a.93@mail.ru (A.T.); gorshkov.sstu@gmail.com (N.G.)

**Abstract:** In this paper, $Li_xCa_{(1-x)}Cu_3Ti_4O_{12}$ (LCCTO) solid solutions were successfully synthesized. XRD diagrams showed that dopant acceptor $Li^+$ cations, in a concentration range of x = 0.01–0.10, were successfully merged into CCTO structure. It was found that doping with low concentrations of lithium (x < 0.05) inhibited grain growth during annealing; however, for x > 0.05, the grain growth process resumed. Permittivity and dielectric losses of obtained LCCTO ceramics were analyzed by the means of impedance spectroscopy in a frequency range from $10^{-1}$ to $10^6$ Hz. It was revealed that acceptor doping with lithium at an appropriate concentration of x = 0.05 allowed to obtain ceramics with a permittivity level of $\varepsilon' = 3 \times 10^4$ and low dielectric losses tanδ < 0.1 at 1 kHz. Further addition of lithium in a concentration range of x = 0.075–0.10 led to a sharp decline in permittivity and an increase in dielectric losses. It was discovered that lithium addition to CCTO ceramics drastically decreased grain boundary resistivity from 115 MΩ·cm to 5–40 MΩ·cm at x = 0.01–0.10. Using Havriliak–Negami equation, the relaxation times for grain dipoles and grain boundary dipoles were found to be ranging from $0.8 \times 10^{-6}$ to $1.7 \times 10^{-6}$ s and from $0.4 \times 10^{-4}$ to $7.1 \times 10^{-4}$ s, respectively. The developed materials can be used in the manufacture of Multilayer Ceramic Capacitors (MLCC) as a dielectric.

**Keywords:** $CaCu_3Ti_4O_{12}$; permittivity; dielectric losses; internal barrier layer capacitance (IBLC); electron-pinned defect dipoles (EPDD); Havriliak-Negami equation; impedance

## 1. Introduction

Ongoing development of various technological equipment implicates increasing demand for materials with high permittivity to be used in different electronic devices, mainly energy storage and energy conversion fields [1–9]. The designing of materials with high permittivity that can outperform existing ones allows for the requirements for device miniaturization to be met, and directly improves the capacity to effectively decrease the size of electronic components. Hence, research into dielectrics has become more relevant in recent years. Known ceramic materials of various chemical composition and structure, in particular lead-based ceramics, $PbZrO_3$-based ceramics, $BaTiO_3$-based ceramics, $SrTiO_3$-based ceramics, $K_{0.5}Na_{0.5}NbO_3$-based ceramics, $BiFeO_3$-based ceramics, $Na_{0.5}Bi_{0.5}TiO_3$-based ceramics, $AgNbO_3$-based ceramics, and $NaNbO_3$-based ceramics, regardless of the chemical nature of the doping element, method, and place of doping, have reached the limit of permittivity [10]. Of the above, as is well-known, materials based on $BaTiO_3$, $SrTiO_3$, and $TiO_2$ are used for dielectric purposes in the modern electronics industry [11–15]. Nevertheless, the permittivity of these materials cannot always be sufficient to meet miniaturization requirements.

In addition to existing dielectric materials, calcium-copper-titanate-based $CaCu_3Ti_4O_{12}$ (CCTO) ceramics are being investigated as promising candidates for these applications due to giant permittivity levels ($\varepsilon'$~$10^4$) [16–18]. CCTO-based ceramics comprise semiconductor grains, the boundaries of which are separated by a thin insulator layer; said combination

leads to high permittivity. This particular structure is described as an internal barrier layer capacitor (IBLC) [19], where semiconductor grains act as capacitor electrodes and grain boundaries act as the capacitor layer. According to the IBLC model, the properties of CCTO-based ceramics can be influenced by improving the electrical properties of semiconductor grains and the insulating phase. However, despite their high permittivity, CCTO-based ceramic materials demonstrate high dielectric losses (tan$\delta \approx 0.2$), which lead to relatively large energy losses in practical use and limit the application range. Therefore, it is essential to reach an optimal balance of permittivity and dielectric losses.

One of the most effective ways to do so is the CCTO structure element doping with various cations and anions [20–28], since the dielectric properties of CCTO-based ceramic depend strongly on the nature, amount, and the method of addition of the dopants. Usually, the additives can be divided into donor and acceptor. Donor dopants are elements with a higher ionic charge than the one of substituted ions, a phenomenon which leads to cation vacancies emerging and vise-versa for the acceptor dopants. In recent years, notable progress in controlling the dielectric properties of CCTO ceramics by doping or co-doping with various cations has been achieved by studies on the effect of Ni, Cd, Sr/Zr, Sr/Mg, and Li. It is shown that a sample of the ceramic material $Ca_{0.9}Sr_{0.1}Cu_{2.9}Mg_{0.1}Ti_4O_{12}$ has the smallest value of the dielectric loss tangent among all other doped compositions (0.05 at a frequency of 1 kHz) [29]. According to the results of [20], the co-doping of Sr and Zr simultaneously increases the permittivity and reduces dielectric losses, and the maximum permittivity (~$2 \times 10^4$) and minimum dielectric losses (~0.8) at a frequency of 1 kHz demonstrates $Ca_{0.8}Sr_{0.2}Cu_3Ti_{4-x}Zr_xO_{12}$ (x = 0.3). Ni-doping (20%) in the study [30] resulted in improved dielectric properties in a CCTO system, namely in a balance of the permittivity (1.51 $\times 10^5$) and dielectric losses (0.051) at room temperature and frequency 1 kHz. When $Li_2CO_3$ is doped in an amount of 0.5 wt.%, the permittivity is maintained at a level of $10^5$ with a weak dependence on frequency below $10^5$ Hz, and its loss tangent decreases below 0.1 in the range from 300 Hz to 5 kHz (with a minimum value of 0.06 at 1 kHz) [31].

The characteristics of CCTO-based ceramics are also extremely sensitive to annealing conditions, the most important of which are temperature and duration [28,32–34]. For instance, obtaining CCTO-based ceramics with high permittivity requires prolonged thermal treatment at 1080 °C, although temperatures above 1065 °C cause $Cu^{2+}$ reduction to $Cu^+$, which, in turn, can induce large dielectric losses. Hence, it is advisable to dope the CCTO structure with additives affecting heavily both the calcining temperatures and the microstructure, e.g., with Li-dopants. Studies [35,36] suggest that the CCTO co-doping with lithium fluoride allows for a ceramic structure to be obtained with decreased dielectric losses while preserving the high dielectric constant level. Particularly, it was reported that ceramics with dielectric losses (tan$\delta$) of ~0.06 and permittivity ($\varepsilon'$) of ~7.7–8.8 $\times 10^4$ were obtained. A significant decline in tan$\delta$ level and superior non-linear characteristics were attributed to a significant rise in grain boundary resistivity, which was linked to a greatly enhanced potential barrier and activation energy of conductivity in grain boundaries. In the article [37] it is shown that co-doping with $Li^+$ and $Al^{3+}$ ions at the $Cu^{2+}$ site can lower the appropriate annealing temperatures and enhance the dielectric characteristics of CCTO-based ceramics. Nevertheless, these reports do not state the influence of lithium ions on the ceramics' properties separately; moreover, the $Li^+$ ions have been used to fill the $Cu^{2+}$ site. In this respect, one can assume that doping with lithium ions at the Ca sites can induce considerable microstructure and dielectric properties changes in CCTO. Therefore, this paper is mainly focused on acceptor doping of CCTO with $Li^+$ cations at $Ca^{2+}$ sites.

It should be noted that the concentration of dopants, which has a positive effect on the structure and functional properties of CCTO ceramics, is at the microlevel and does not exceed 0.5 mol.%. According to some individual studies, in the case of Li-doping, this value decreases to 0.1 mol.% due to the formation of impurity crystalline phases in the phase composition of the synthesized material at higher concentrations. Additionally, the reverse effect of grain growth is observed instead of the desired growth inhibition [31,38]. The simplest and most accessible method of synthesis, which makes it possible to control the

stoichiometry of precursors to obtain single-phase ceramics with microamounts of dopants, is the solid-phase reaction.

This paper's aim is to investigate the structural and dielectric properties of Li-doped $Li_xCa_{(1-x)}Cu_3Ti_4O_{12}$-ceramics (x = 0.010; 0.025; 0.050; 0.075; 0.100) obtained by solid-state method.

## 2. Materials and Methods

### 2.1. Synthesis

The Li-doped CCTO ceramic powders were obtained via solid-state synthesis using titanium dioxide ($TiO_2$, rutile, 99.5%, «Component-Reaktiv», Moscow, Russia), copper (II) oxide (CuO, 99%, «Vekton», St. Petersburg, Russia), calcium carbonate ($CaCO_3$, «AO Reahim», Moscow, Russia), and lithium carbonate ($Li_2CO_3$, 99%, «Rushim», Moscow, Russia) powders. Stoichiometric amounts of the abovementioned powders were calculated using the $Li_xCa_{(1-x)}Cu_3Ti_4O_{12}$ (x = 0.010; 0.025; 0.050; 0.075; 0.100) formula and grinded in a high-energy mill (Fritsch Pulverisette 9) in ethanol media with $Si_3N_4$ beads. Resulting oxide mixtures were thermally treated in a muffle furnace at 900 °C for 8 h with subsequent deagglomeration in the high-energy Fritsch Pulverisette 9 mill.

Dielectric properties of prepared samples were studied, firstly, by single-axis pressing of the powders into ~12 mm-diameter discs (~1–1.5 mm thick) at 100 MPa. Polyvinyl alcohol (PVA) was used as a binder (5 wt.%) The obtained discs were sintered in a muffle furnace at 1050 °C for 8 h on an $Al_2O_3$-substrate in air. The sintered polished discs surfaces were painted with silver paste and cured at 690 °C for 1 h.

### 2.2. Sample Investigation

Phase analysis of prepared samples was carried out by means of X-ray phase diffraction (XRD) (ARL X'TRA, Thermo Scientific, Ecublens, Switzerland) using CuKα-radiation (λ = 0.15412 nm) in the angle range of 2θ from 20 to 80 degrees. The microstructure of powders and the fractured discs of ceramics, without additional treatment, was studied via scanning electron microscopy (SEM) (Aspex EXplorer, Aspex LLC, Framingham, MA, USA). The dielectric properties were investigated by means of impedance spectroscopy (Novocontrol Technologies GmbH & Co. KG, Montabaur, Germany) in a frequency range from $10^{-1}$–$10^6$ Hz at room temperature (excitation amplitude of 50 mV).

## 3. Results and Discussion

XRD patterns of basic and doped CCTO powders and sintered discs based on them are shown in Figure 1a,b. The main peaks correspond to body-centered cubic lattice of the $CaCu_3Ti_4O_{12}$ structure (Im3) listed in a standard JCPDF#75-2188 database. It should be noted that the absence of secondary phases (such as CaO and $Li_2O$) demonstrates that the $Li^+$ ions are embedded into the CCTO crystal lattice in listed quantities. The diffraction pattern of the powders revealed reflections (2θ = 36 and 74°) of the impurity phase of CuO. However, the intensity of these reflections was very low; therefore, the presence of the oxide phase is insignificant.

The phase compositions of sintered discs after polishing include three crystalline phases. The main phase of ceramics remains $CaCu_3Ti_4O_{12}$; however, in addition to CuO, trace phases of $TiO_2$ are also observed for the sintered samples, while no phases involving Li are discernible in any of the XRD patterns, regardless of dopant concentration.

SEM images of $Li_xCa_{(1-x)}Cu_3Ti_4O_{12}$ powders with various $Li^+$ concentrations are shown in Figure 2. The morphology of the samples prepared at 900 °C comprises homogenous irregular spheres. The size of the spheres is estimated at ~2 μm and is not influenced by the $Li^+$ ions concentration.

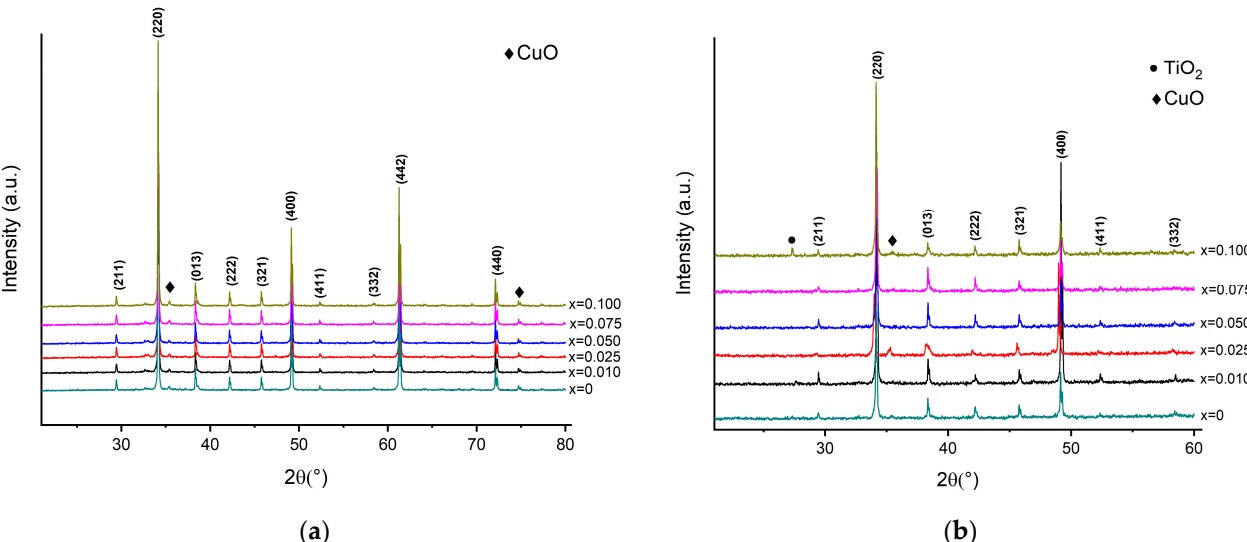

**Figure 1.** XRD patterns of basic CCTO and doped $Li_xCa_{(1-x)}Cu_3Ti_4O_{12}$ (**a**) powders annealed at 900 °C and (**b**) discs sintered at 1050 °C.

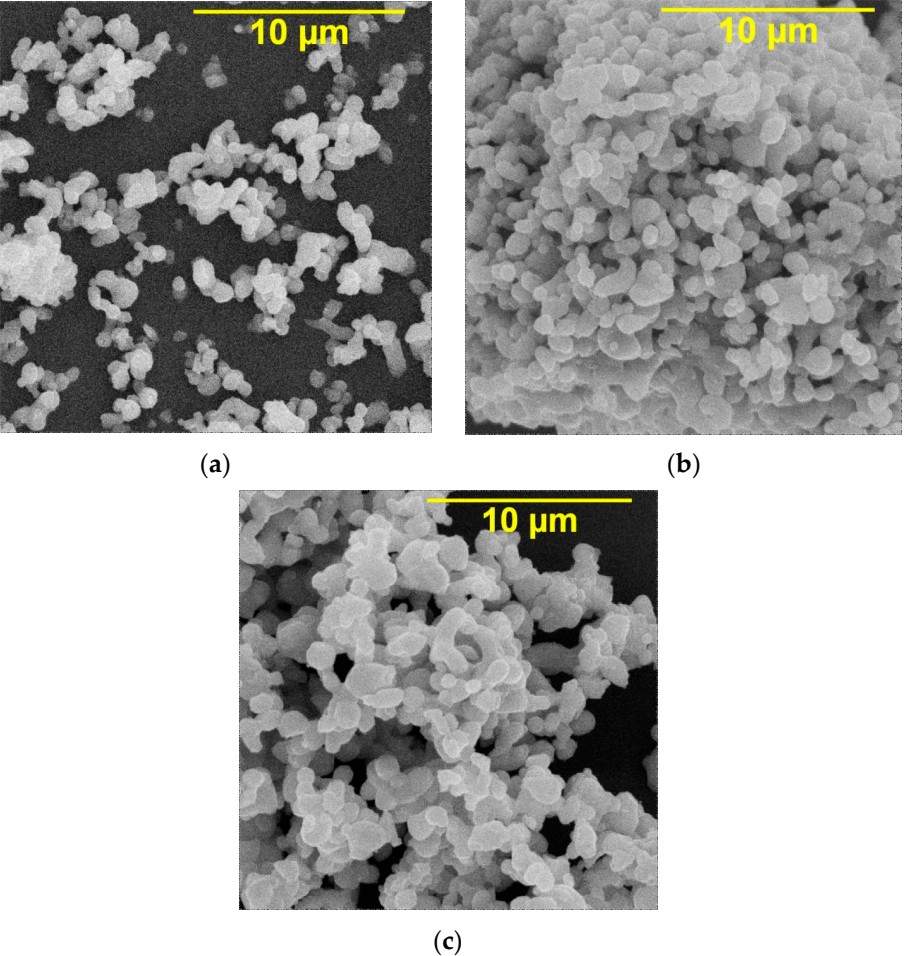

**Figure 2.** SEM images of $Li_xCa_{(1-x)}Cu_3Ti_4O_{12}$ powders (**a**) x = 0; (**b**) x = 0.05; (**c**) x = 0.10.

The cross-section (fractured discs) morphologies of LCCTO ceramics with various $Li^+$ concentrations are shown in Figure 3. It can be noticed that the basic CCTO exhibits a rapid grain growth process up to 20–40 μm size. Additionally, a secondary phase, presumably CuO, present on the grain boundaries can be observed. Doping the CCTO structure with

Li$^+$ ions can strongly influence the microstructure and grain sizes in ceramics. Namely, for x = 0.025, the number of small grains (under 10 μm) is higher; moreover, the maximum grain size decreases to 30 μm. In the case of the concentration being raised to x = 0.05, a further reduction in the size occurs. In this case, the microstructure consists of large grains (up to 20 μm) which are isolated by numerous small grains (2–4 μm). Also, the presence of secondary phases on the grain boundaries becomes less noticeable, which can be evidence of a lower diffusion capacity of copper leading to reduction in grain size and the densification enhancement. As for further addition of dopant (x = 0.075–0.100), the result is the opposite. The grain size and the number of larger grains is significantly expanded. In other words, the doping of Li$_x$Ca$_{(1-x)}$Cu$_3$Ti$_4$O$_{12}$ ceramics can be beneficial to a reduction in grain growth only in case of Li$^+$ ions concentrations being lower than x = 0.05.

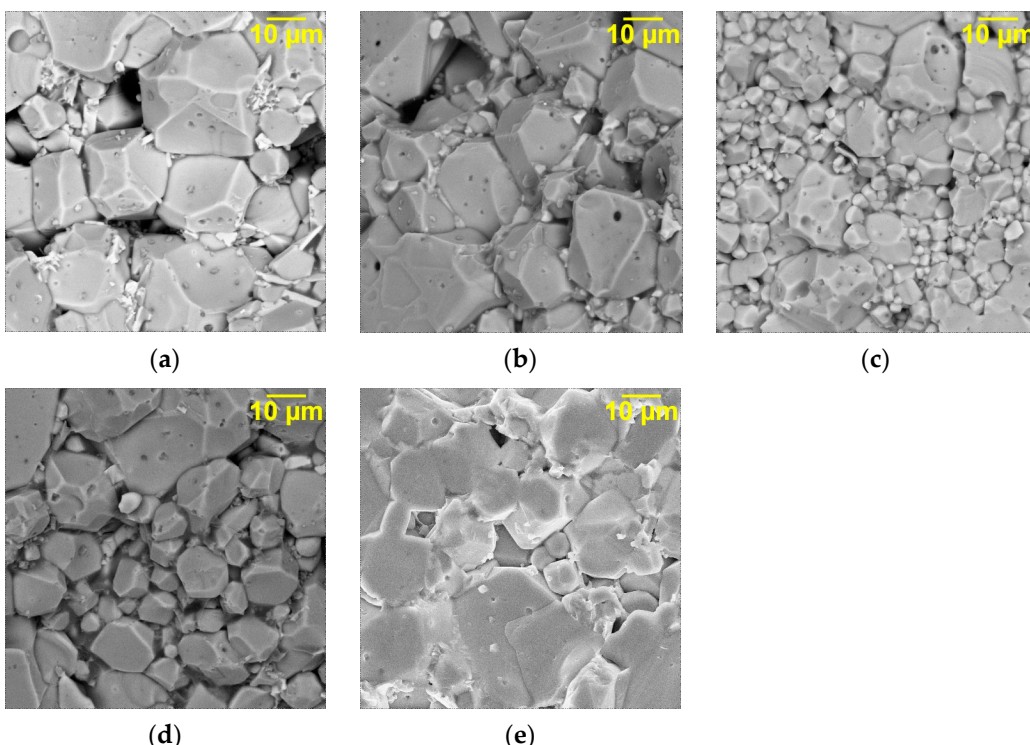

**Figure 3.** SEM images of the morphology of cross-section LCCTO ceramics (**a**) x = 0; (**b**) x = 0.025; (**c**) x = 0.05; (**d**) x = 0.075; (**e**) x = 0.10.

The permittivity and dielectric losses frequency dependencies are shown in Figure 4. As it can be noticed for the permittivity, the basic CCTO and lithium-doped LCCTO (x = 0.01–0.05) show a plateau in the range between 10 Hz and 10 kHz, which is followed by a sharp cut of permittivity values at increasing frequency, attributed to a Debye-like relaxation process [39]. This behavior is typical for the IBLC structure, where a steep decline in permittivity occurs in case of the mean free electron path induced by outer electric field being less than the average grain size at higher frequencies. As a result, the main contribution to high levels of permittivity at higher frequencies is due to the volume of a semiconductor grain. As for the lower frequencies, it can be attributed to the insulator grain boundaries. In addition, the increase in permittivity at lower frequencies (under 10 Hz) occurs due to the charge carriers being accumulated on the grain–boundary interface, which leads to Maxwell–Wagner polarization owing to space charge storage [40]. As the Li$^+$ concentration increases, the dielectric relaxation is present at ~10 Hz, resulting in a sharp drop of permittivity at frequencies higher than 10 Hz. Moreover, the Li doping of CCTO for x = 0.01 and 0.025 promotes the rise in permittivity for frequencies between 1 and 10$^4$ Hz; at the same time, for x = 0.05, it is marked by a slight reduction in permittivity. Notably, the

permittivity value still increases at low concentrations of the dopant despite the shrinking grain size [41]. Thus, higher permittivity can be attributed to growing grain boundary capacitance $C_{gb}$ and surface charge storage occurring on grain boundaries. The basic CCTO has a permittivity of $3.2 \times 10^4$, while samples with x = 0.01; 0.025; 0.05 are characterized by permittivity values of $5.5 \times 10^4$; $4.5 \times 10^4$; $2.8 \times 10^4$, respectively, at 10 kHz. Further addition of Li$^+$ leads to a sharp cut in permittivity in the $10–10^6$ Hz range. Low permittivity can also be attributed to the fact that the sintering temperature of 1050 °C is insufficient for the formation of insulator grain boundary layer for these Li$^+$ concentrations. Such an effect is exhibited presumably owing to the high CuO deficiency on the grain boundaries [42]. Only a small number of Cu ions undergo the oxidation–reduction reaction due to the oxygen vacancies present for the optimal charge balance in case of CCTO ceramics being sintered at elevated temperatures. Ultimately, this leads to the formation of $Cu^{2+}$ and $Ti^{3+}$ which determine the high permittivity. The abovementioned reactions typically occur above 1065 °C in basic CCTO ceramics [19]. Hence, in case of lower Li$^+$ concentrations (x = 0.01; 0.025), the high permittivity level can be attributed to a high content of CuO segregated along the grain boundaries, while the increase to x = 0.05 results in copper deficiency which promotes the oxygen diffusion along grain boundaries, thus making up for the excessive in-grain charge [43–45]. Furthermore, acceptor doping CCTO structures with lithium cations can lead to changes in the oxidation state ($Cu^+/Cu^{2+}$; $Ti^{3+}/Ti^{4+}$) for charge compensation, which [46–48], in turn, results in a higher number of crystal dipoles, additional attribution to polarization, and, therefore, in the increase of permittivity.

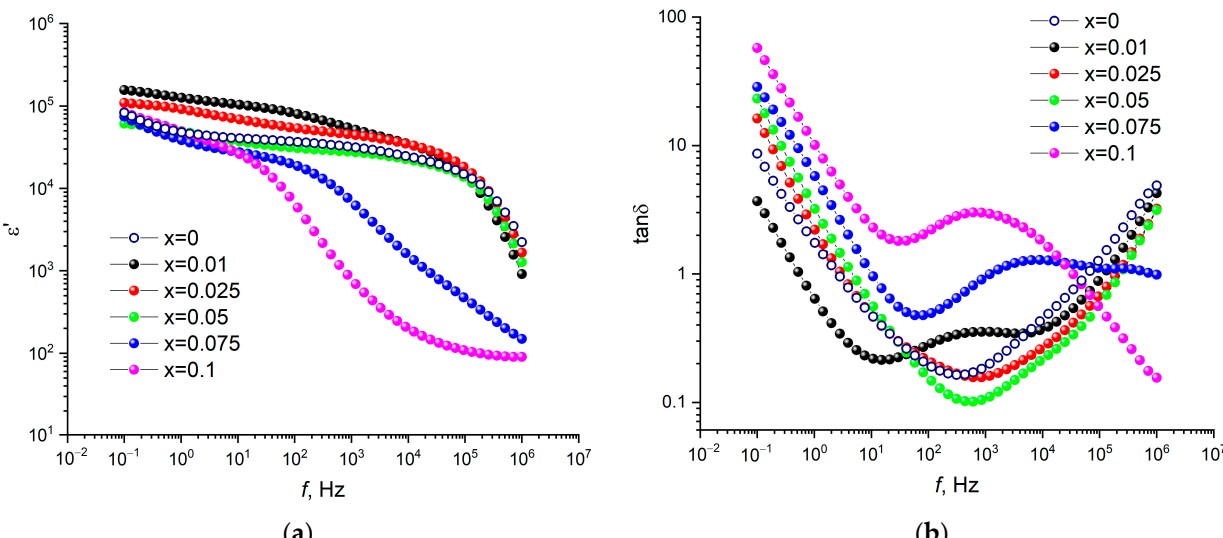

**Figure 4.** Frequency dependence of (**a**) dielectric constant and (**b**) dielectric loss of Li$_x$Ca$_{(1−x)}$Cu$_3$Ti$_4$O$_{12}$ ceramics.

As it can be evinced from Figure 4b, basic and doped CCTO ceramics show the lowest dielectric losses (tanδ) for x = 0.025 and x = 0.05, both of which tend to decline with increasing frequency from 1 to 10–1000 Hz and then rapidly grow in the higher frequency range. This trend is present due to the semiconductor nature of the grain which has a relatively higher permittivity value than the one of the insulator grain boundaries. In this case, the addition of x = 0.025 and 0.05 lithium concentrations induces the drop in the minimal value of dielectric losses to 0.15 and 0.10, respectively, as opposed to basic CCTO having tanδ of 0.18 at 1 kHz. Similarly to permittivity, the dielectric losses show a sharp rise when further doped with Li$^+$ (x > 0.025). This can be a result of the poor insulation properties of grain boundaries, leading to high leakage current. Additionally, the decrease in dielectric losses at concentrations x < 0.05 can be attributed to the shrinkage in grain sizes, leading to growing number of insulator grain boundary layers [23,49].

In order to analyze the grain boundary resistivity, the complex impedance spectroscopy diagrams Z″-Z′ (Figure 5a) were studied. Impedance spectra comprise a semicircle for frequencies under 10 kHz and a semicircle for higher frequencies, which is evidence of extremely low resistivity. As it can be observed, the $Ca^{2+}$ substitution with $Li^+$ in CCTO lowers the specific resistivity estimated from the semicircle intersection with the Z′ point. According to the IBLC model, the high-frequency semicircle corresponds to grain contribution and exhibits extremely low resistivity values, whereas the low-frequency semicircle determines the grain boundary contribution and shows great specific resistivity. The equivalent scheme for these type of impedance spectra consists of two serially connected RC elements, as shown in Figure 5b where $R_b$ and $R_{gb}$ are specific resistivities of grains and grain boundaries, respectively, and CPE1 and CPE2 determine the grains and grain boundaries permittivity. Based on the scheme and complex impedance spectra, the values of grain boundaries specific resistivity $R_{gb}$ were estimated using the EIS Spectrum Analyser software, as shown in Figure 5b. As it can be seen, introducing a small amount of $Li^+$ dopant sharply reduces the $R_{gb}$, which gradually continues to decrease with further $Li^+$ addition. Generally, the lowering in grain size results in growing grain boundary resistivity $R_{gb}$; however, in this case, it is accompanied by a decline in the $R_{gb}$ value. Thus, one can assume that the decrease in $R_{gb}$ is mainly caused by declining segregation on copper at the grain boundaries.

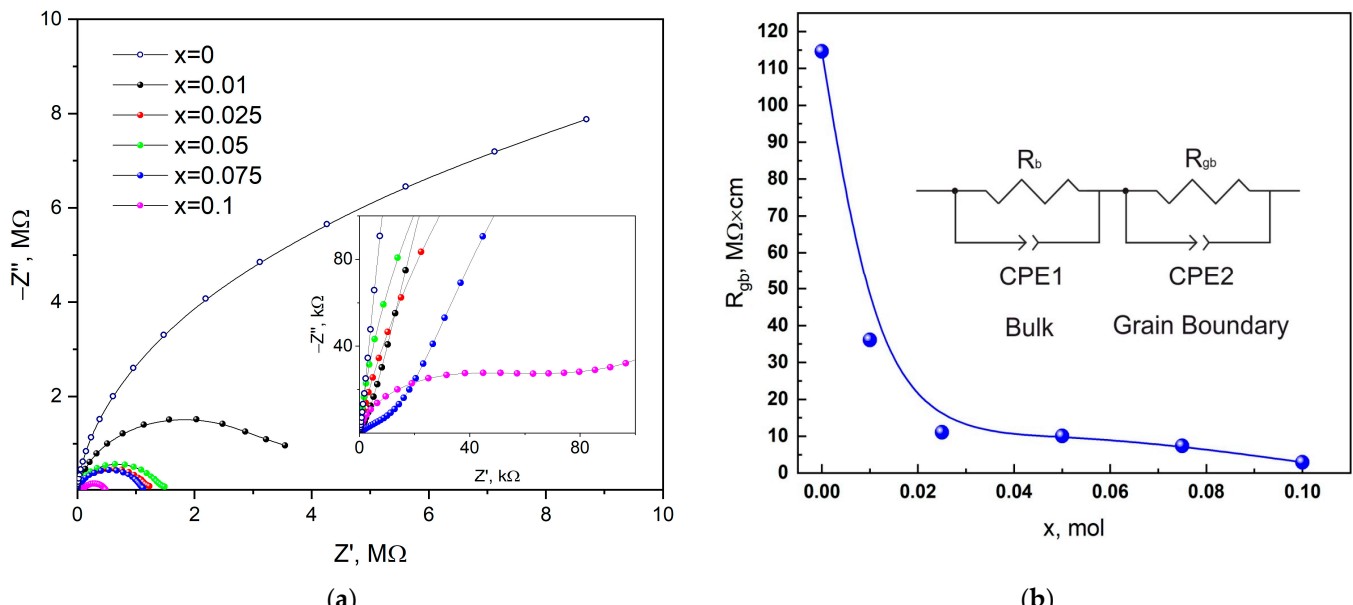

(a)                                                                    (b)

**Figure 5.** (**a**) The impedance spectra for LCCTO ceramics modified by different amounts of Li+ at room temperature. (**b**) Dependence of the grain boundary resistance $R_{gb}$ on the $Li^+$ cation concentration.

The Cole–Cole diagram for LCCTO is shown in Figure 6. As it can be observed, the graphs show two semicircles which are attributed to the grain and boundary influence. The third semicircle is present on the x = 0.01 sample diagram due to excessive contribution of the ceramic–electrode interface. At the same time, the grain semicircle is present in the high-frequency range (closer to the origin), followed by the boundary semicircles. Meanwhile, they display well-split relaxation times. Additionally, the doping of CCTO ceramics with different concentrations of the additive can lead to the change in dipole relaxation times. The contribution of induced dipole grain orientation (electron-pinned defect-dipole (EPDD)), grain boundaries (internal barrier layer capacitor (IBLC)), and ceramic–electrode

interface [50,51], as well as the parameters of the corresponding relaxations, can be estimated using the Havriliak–Negami (HN) function:

$$\varepsilon^* = \frac{j\sigma_{DC}}{\omega\varepsilon_0} + \left(\varepsilon_\infty + \frac{\Delta\varepsilon}{\left(1 + (j\omega\tau)^\alpha\right)^\beta}\right)_{EPDD} + \left(\varepsilon_\infty + \frac{\Delta\varepsilon}{\left(1 + (j\omega\tau)^\alpha\right)^\beta}\right)_{IBLC} \tag{1}$$

where $\varepsilon^*$ is the complex permittivity; $\omega$ is the angular frequency of the alternating electric field; $\tau$ is the average relaxation time; $\sigma_{DC}$ is the direct current conductivity; $\alpha$ is the parameter characterizing the dispersion width of the relaxation time; and $\beta$ is the peak of the parameter asymmetry, $\Delta\varepsilon = \varepsilon_s - \varepsilon_\infty$. Here, $\varepsilon_s$ is the electrostatic permittivity for low frequencies, while $\varepsilon_\infty$ is the permittivity for high frequencies.

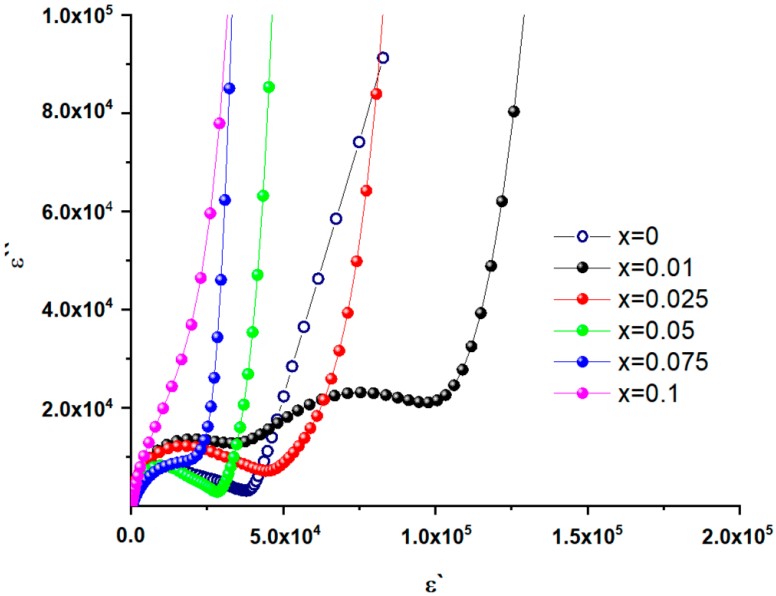

**Figure 6.** Measured complex dielectric Cole–Cole plot.

HN function parameters for LCCTO ceramics estimated using DielParamFit_2 [52] software are shown in Table 1. As shown in Table 1, the relaxation behavior of the grain dipoles obeys the Cole–Cole function ($\alpha$ = 0.88–0.94, $\beta$ = 1). In other words, the grain dipoles comprise a combination of relaxation time with a narrow distribution, as the $\alpha$ parameter is close to the value of 1 in all of the cases. The estimated grain dipole relaxation time ranges from $0.78 \times 10^{-6}$ to $1.70 \times 10^{-6}$ s. The dipole behavior of the grain boundaries also obeys the Cole–Cole function ($\alpha$ = 0.45–0.57, $\beta$ = 1); however, the $\alpha$ parameter is much less than 1. This indicates the presence of an interacting dipole combination on the grain boundaries which is characterized by a relaxation time dispersion. The estimated grain boundary dipole relaxation time ranges from $0.4 \times 10^{-4}$ to $7.1 \times 10^{-4}$ s.

**Table 1.** Parameters of the Havrilyak-Negami equation for LCCTO ceramics.

| Parameters of the Havriliak-Negami Equation | x = 0 | | x = 0.01 | | x = 0.025 | | x = 0.05 | |
|---|---|---|---|---|---|---|---|---|
| | EPDD | IBLC | EPDD | IBLC | EPDD | IBLC | EPDD | IBLC |
| $\Delta\varepsilon$ | 15,846 | 2549 | 26,950 | 92,835 | 31,780 | 40,261 | 16,808 | 16,256 |
| $\alpha$ | 0.88 | 0.45 | 0.93 | 0.57 | 0.84 | 0.57 | 0.94 | 0.50 |
| $\beta$ | 1 | 1 | 1 | 1 | 1 | 1 | 1 | 1 |
| $\tau, s$ | $0.8 \times 10^{-6}$ | $0.6 \times 10^{-4}$ | $1.7 \times 10^{-6}$ | $6.2 \times 10^{-4}$ | $1.6 \times 10^{-6}$ | $7.1 \times 10^{-4}$ | $0.9 \times 10^{-6}$ | $0.4 \times 10^{-4}$ |

At the same time, it should be noted that for concentrations of 0.01 and 0.025 an increase in the dipole component of polarization to 15,846 and 31,780, respectively, is observed, with a slight change in the parameters $\alpha$ and $\beta$ relative to the undoped CCTO. The relaxation time for EPDD increases to $1.6 \times 10^{-6}$ s, which is probably due to the polaron conductivity of the CCTO; therefore, doping with small amounts of mobile lithium leads to an increase in the permittivity due to this process. The contribution of the barrier layer in the studied system, similarly to the dipole component, increases to x = 0.025 and sharply decreases at x = 0.05. The calculated $\Delta\varepsilon$ increases from 2549 to 40,261 for x = 0.025 relative to the undoped CCTO. This behavior is related to the ratio of grain sizes and the area of grain boundaries, as well as to a change in the chemical and phase composition. This is confirmed by results of SEM and XRD analyses. Such an influence is complex, and a clear improvement in the dielectric properties is observed for concentrations of x = 0.01 and 0.025.

The developed materials can be used in the manufacture of MLCC as a dielectric.

## 4. Conclusions

In this paper, the authors successfully obtained $Li_xCa_{(1-x)}Cu_3Ti_4O_{12}$ (LCCTO) ceramics following a solid-state reaction at 900 °C. XRD data show that the acceptor dopant $Li^+$ ions in a x = 0.01–0.10 content range were successfully merged into the CCTO lattice structure. It was found that doping with low lithium concentrations (x < 0.05) inhibited grain growth during the ceramics annealing process; however, the growth proceeded for lithium content x > 0.05. The permittivity and the dielectric losses were studied by means of impedance spectroscopy of the obtained LCCTO ceramic samples. Doping the CCTO structure with $Li^+$ in the x = 0.01–0.025 content range facilitated a minor increase in permittivity in the $1$–$10^4$ Hz frequency range, whereas for x = 0.05 permittivity was found to be slightly lower. Additionally, for x = 0.025 and 0.05, a decline in dielectric losses at the extreme was observed (0.15 and 0.10, respectively), compared to 0.18 for basic CCTO at 1 kHz. Further addition of the dopant up to x = 0.075–0.10 resulted in a sharp decline in permittivity and in dielectric losses growth, which can be attributed to a CuO deficiency along the grains. The Nyquist plot $Z''$-$Z'$ allowed for the grain boundary resistivity $R_{gb}$ to be estimated. It was found that lithium addition to the CCTO ceramics drastically reduced the grain resistivity from 115 MΩ·cm to 5–40 MΩ·cm for x = 0.01–0.10. The Cole–Cole diagram $\varepsilon''$-$\varepsilon'$ analysis using the Havriliak–Negami equation allowed for grain dipole and grain boundary dipole relaxation times to be estimated. Mean grain dipole relaxation time ranged from $0.8 \times 10^{-6}$ to $1.7 \times 10^{-6}$ s, while the estimated relaxation time of the grain boundary dipole combination was found to be ranging from $0.4 \times 10^{-4}$ to $7.1 \times 10^{-4}$ s. The obtained results can be used for developing high-permittivity materials, e.g., for ceramic capacitor applications. The developed materials can be used in the manufacture of MLCC as a dielectric.

**Author Contributions:** Conceptualization, A.T. and N.G.; methodology, A.T., N.M. and A.A.; software, D.A. and I.Z.; validation, M.V. and A.G.; formal analysis, A.T. and N.G.; investigation, N.M. and A.A.; resources, N.G. and A.G.; data curation, A.G. and N.G.; writing—original draft preparation, A.T. and M.V.; writing—review and editing, A.G. and N.G.; visualization, D.A., I.Z. and N.G.; supervision, A.T.; project administration, N.G.; funding acquisition, A.G. All authors have read and agreed to the published version of the manuscript.

**Funding:** This research was funded by the Russian Science Foundation, grant number 19-73-10133, https://rscf.ru/en/project/19-73-10133/ (accessed on 3 August 2022).

**Data Availability Statement:** Not applicable.

**Conflicts of Interest:** The authors declare no conflict of interest. The funders had no role in the design of the study; in the collection, analyses, or interpretation of data; in the writing of the manuscript, or in the decision to publish the results.

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
