# Peer review of "The Effect of Lithium Doping on the Dielectric Properties of Solid Solutions LixCa(1−x)Cu3Ti4O12 (x = 0.01–0.1)"

_jcs, doi:10.3390/jcs7070282_

Round 1

Reviewer 1 Report

Follow my notes in the attached pdf. Only minor items are there indicated. I am generally satisfied with the article content, if I am accepting the doubtful idea that Li is useful for these materials with giant permittivity. 

Good. Avoid the use of "and etc" - this is the only systematic error.

Author Response

Point 1: You should say more - why exactly these concentrations are used. - Why the solid state method is used.

Response 1: Thank you for your comment. More information about selected concentration range and synthesis method was added to introduction. (in red)

Point 2: Better said SINTERED, since they were not rigid enough before this step.

Response 2: Thank you for your comment. It was corrected. (in red)

Point 3: How obtained. Sectioned + polished? Or "fractured". Or how exactly?

Response 3: Thank you for your comment. Information about it was added in section of 2.2 Sample investigation. (in red)

Point 4: Li ion in really small one and its mobility is high in this way. - I have let say also the general doubts, if Li is reasonable dopant for CCTO.

Response 4: Thank you for your comment. Discussion about role Li ion as dopant for CCTO ceramics was improved in Introduction. In section of Results and Discussion more comments were added. (in red)

Reviewer 2 Report

Dear authors, please find the following comments and suggestions to your manuscript.

1.      In the Experimental part - Was TiO2 in anatase modification from Komponent-Reactiv? From the own experience, they produced only the rutile modification. Also, please prove that Rushim is a Chinese company but not a reagent supplier from Moscow.

2.      The authors didn’t mention any procedures for the preparation of the ceramic samples to the dielectric measurements except to their manufacturing from the powders by pressing and annealing. Usually, the surfaces of the ceramic samples are polished and tested for plane parallelism in order to meet the measurements conditions. Otherwise the painted electrodes appear not flat and the measurements are unreliable.

3.      XRD patterns in Figure 1 demonstrated the peaks of CCTO phase marked with the relevant Miller indices. However, there are several side peaks which received no description neither in the figure nor in the text. Moreover, the authors claimed that “It can be noted that the absence of secondary phases…” So, it remained unclear what that peaks were attributed to? Please complete the phase analysis.

4.      In Figure 3, the secondary phase found at the grain boundaries should be highlighted.

5.      XRD analysis of the manufactured ceramic samples must be provided because their phase contents were obviously different from the synthesized powders. That is because in the lines 110-111 the authors noted the absence of CuO, CaO, Li2O phases in the powders, but in line 124 and further discussed the presence of CuO phase in the ceramics. Besides, a comparison of the cell parameters in CCTO and LCCTO powders as well as in the corresponding ceramics would illustrate the incorporation of lithium ions in the structure.

6.      Could the authors provide any explanation for the peculiarities in the grain growth at different lithium concentration? From the current version of the manuscript, it’s not clear.

Minor grammar chech is required.

Author Response

Point 1: In the Experimental part - Was TiO2 in anatase modification from Komponent-Reactiv? From the own experience, they produced only the rutile modification. Also, please prove that Rushim is a Chinese company but not a reagent supplier from Moscow.

Response 1: Thank you for your comment. It was corrected. (in red)

Point 2: The authors didn’t mention any procedures for the preparation of the ceramic samples to the dielectric measurements except to their manufacturing from the powders by pressing and annealing. Usually, the surfaces of the ceramic samples are polished and tested for plane parallelism in order to meet the measurements conditions. Otherwise the painted electrodes appear not flat and the measurements are unreliable.

Response 2: Thank you for your comment. It was corrected. (in red)

Point 3: XRD patterns in Figure 1 demonstrated the peaks of CCTO phase marked with the relevant Miller indices. However, there are several side peaks which received no description neither in the figure nor in the text. Moreover, the authors claimed that “It can be noted that the absence of secondary phases…” So, it remained unclear what that peaks were attributed to? Please complete the phase analysis.

Response 3: Thank you for your comment. XRD studies were added and corrected.

Point 4: In Figure 3, the secondary phase found at the grain boundaries should be highlighted.

Response 4: Thank you for your comment. It was corrected.

Point 5: XRD analysis of the manufactured ceramic samples must be provided because their phase contents were obviously different from the synthesized powders. That is because in the lines 110-111 the authors noted the absence of CuO, CaO, Li2O phases in the powders, but in line 124 and further discussed the presence of CuO phase in the ceramics. Besides, a comparison of the cell parameters in CCTO and LCCTO powders as well as in the corresponding ceramics would illustrate the incorporation of lithium ions in the structure.

Response 5: Thank you for your comment. Figure 1b was adedd with its discussion. (in red)

Point 6: Could the authors provide any explanation for the peculiarities in the grain growth at different lithium concentration? From the current version of the manuscript, it’s not clear.

Response 6: Thank you for your comment. Yes, you are right, influence of lithium on grain growth of CCTO ceramics is complex. We tried to discuss this process in finished part of manuscript. (in red)

Round 2

Reviewer 1 Report

See attached file. Just few very formal items indicated.

The manuscript is nearly ready to publish.

Author Response

Point 1: You mentioned multi-layer co-fired ceramics? You should explain this with using HERE the full words.

Response 1: Thank you for your comment. It was corrected as Multilayer Ceramic Capacitors. (in red)

Reviewer 2 Report

Dear authors,

Thank you for the provided corrections.

However, there are some points in the phase analysis which require your attention. Please compare the figures 1a and 1b. In the first of them, you marked Miller indices of CCTO, including (411) and (332). In Figure 1b, you skipped the (332), even though it is visible at least for some of the samples. The peak which was attributed to CCTO (411) in fig. 1a somehow transformed into a peak of CuO. Please clarify this. Besides, there is quite a pronounced peak at about 43-44 deg. 2theta in fig. 1b, that you didn's explain.

In the relevant text you claimed, that the prexence of CuO in the powder was insignificant (lines 142-143). But it's presence both in powder and ceramics is unclear, as you used stoicheometric mixtures in the synthesis. Taking into account the presence of TiO2 traces in the powder as well as volatility of lithium component, one can conclude that real concentrations of lithium ions in the prepared solid solutions are different from the theoretical. Sure you understand the importance of real contents of the solid solutions for the further study of their dielectric properties that you provided. Please determine the real lithium contents in your ceramic samples.

Grammar check is required.

Author Response

Point 1: However, there are some points in the phase analysis which require your attention. Please compare the figures 1a and 1b. In the first of them, you marked Miller indices of CCTO, including (411) and (332). In Figure 1b, you skipped the (332), even though it is visible at least for some of the samples. The peak which was attributed to CCTO (411) in fig. 1a somehow transformed into a peak of CuO. Please clarify this. Besides, there is quite a pronounced peak at about 43-44 deg. 2theta in fig. 1b, that you didn's explain.

Response 1: Thank you for your comment. It was corrected. Peak at about 43-44 deg. 2theta in Fig. 1b is device error (namely, empty cuvette signal). It was due to the study of tablets, not powders. It was corrected.

Point 2: In the relevant text you claimed, that the prexence of CuO in the powder was insignificant (lines 142-143). But it's presence both in powder and ceramics is unclear, as you used stoicheometric mixtures in the synthesis. Taking into account the presence of TiO2 traces in the powder as well as volatility of lithium component, one can conclude that real concentrations of lithium ions in the prepared solid solutions are different from the theoretical. Sure you understand the importance of real contents of the solid solutions for the further study of their dielectric properties that you provided. Please determine the real lithium contents in your ceramic samples.

Response 2: Thank you for your comment. The difficulty of working with such a mobile and volatile element as lithium is accompanied by equipment limitations in identifying it, since it is a very light element. Unfortunately, we cannot supplement the manuscript with an analysis of the lithium content in the composition of ceramic samples.

Round 3

Reviewer 2 Report

The paper could be accepted in the present form.